

# Happiness, depression, physical activity and cognition among the middle and old-aged population in China: a conditional process analysis

Xiaojuan Shi[1], Xiaoxue He[1], Degong Pan[1], Hui Qiao[1] and Jiangping Li[1,2]

[1] Department of Epidemiology and Health Statistics, School of Public Health and Management, Ningxia Medical University, Yinchuan, China
[2] Key Laboratory of Environmental Factors and Chronic Disease Control, Ningxia Medical University, Yinchuan, China

## ABSTRACT

**Background**. Happiness is one variable of subjective well-being, which has been increasingly shown to have protective effects on health. Although the association between happiness and cognition has been established, the mechanism by which happiness leads to cognition remains unclear. Since happiness, depression, and physical activity may all be related to cognition, and happiness is related to depression and physical activity, this study explored the effect of depression and physical activity on the relationship between happiness and cognition among middle and old-aged individuals in China.

**Methods**. Data on 14,344 participants above 45 years of age were obtained from the 2018 China Family Panel Studies survey. A multiple linear regression analysis was performed to identify the correlation factors of cognition. The conditional process analysis was used to assess the mediatory effect of depression and physical activity on the relationship between happiness and cognition.

**Results**. Residence, age, sex, income level, social status, smoking, napping, reading, education, exercise times, satisfaction, happiness, and depression had associations with cognition. When other variables were held constant, cognition score increased by 0.029 standard deviation(SD) for every 1 SD increased in happiness. Mediation analysis showed that happiness had a significant positive total effect on cognition. The direct effect of happiness was significant and accounted for 57.86% of the total effect. The mediatory effect of depression (path of happiness→depression→cognition) accounted for 38.31% of the total effect, whereas that of physical activity (path of happiness→exercise times→cognition) accounted for 3.02% of the total effect.

**Conclusion**. Happiness has a positive correlation with cognitive function, and depression and physical activity play mediatory roles in this association. Effective interventions to improve happiness levels of middle and old-aged population will not only improve their subjective well-being but also improve their cognitive function, which carries great potential for reducing public health burdens related to cognitive aging.

Corresponding authors
Hui Qiao, qiaohui71@163.com
Jiangping Li, lijp@nxmu.edu.cn

## INTRODUCTION

Cognition is the most complex function of the brain. Cognitive decline is a significant concern among middle and old-aged individuals, particularly in China, a country with rapidly aging population. Age-related cognitive changes are cumulative irreversible changes occurring in a long-term process (*Li, Huxhold & Schmiedek, 2004*). According to studies on cognitive aging, with increasing age, individuals generally have cognitive decline, which affects many cognitive domains, including memory attention, reasoning, and executive function (*Park et al., 1996*; *Salthouse, 2010*; *Schaie, Willis & Caskie, 2004*). The decline in cognitive ability is related to impaired daily functioning among the elderly; however, severity varies among different individuals. Therefore, knowledge of factors that prevent cognitive impairment in old age is necessary and should be elucidated.

Studies have shown that besides age, which is an irreversible factor, many factors are related to changes in human cognitive function. Individuals are often concurrently exposed to both risk and protective factors, which may interact with each other during the life course of individuals, and the net effect of these interactions determines the overall risk of cognitive impairment. Educational level (*Lövdén et al., 2020*), sleep (*Dzierzewski, Dautovich & Ravyts, 2018*), physical activity (*Gheysen et al., 2018*), smoking (*Amini, Sahli & Ganai, 2021*), depression (*Shimada et al., 2014*), malnutrition (*Corish & Bardon, 2019*), and happiness (*Tan et al., 2019*) may all be linked to cognitive decline. Among these factors, the effect of happiness on cognitive function deserves added attention because the potential mechanisms of action are likely distinct from those of other interventions in improving cognition.

Happiness, a variable of subjective well-being, is defined as "the overall appreciation of one's life-as-a-whole" (*Danner, Snowdon & Friesen, 2001*). Happiness confers several protective effects on health. Happy individuals tend to live longer and be healthier and more resilient (*Danner, Snowdon & Friesen, 2001*; *Fredrickson, 2003*; *Veenhoven, 2008*). According to previous studies, individuals with low subjective well-being are more likely to experience cognitive decline and dementia (*Boyle et al., 2010*; *Sutin, Stephan & Terracciano, 2018*). Despite the known associations between happiness and cognitive function, the potential mechanisms underlying these associations are currently poorly understood. A better understanding of how happiness is associated with cognition is crucial, given the beneficial effects of happiness on health and the deleterious effect of cognitive impairment on life.

Older individuals with higher levels of happiness are more likely to be physically active (*Kim, Strecher & Ryff, 2014*; *Steptoe, 2019*) and live healthier and longer (*Veenhoven, 2008*). Many longitudinal studies have shown that physical activity reduces cognitive decline and the risk of dementia (*Blondell, Hammersley-Mather & Veerman, 2014*; *Carvalho et al., 2014*; *Fratiglioni, Paillard-Borg & Winblad, 2004*). Happiness is a positive psychological state, whereas depression is a negative psychological state. According to several reports, individuals with depression tend to have more severe cognitive impairment (*Kim et al., 2016*; *Wong et al., 2015*). A cohort study with over three years of follow-up in Spain revealed

that depression significantly increased the risk of mild cognitive impairment among older adults (*Lara et al., 2017*).

Since happiness, depression, and physical activity may all be related to cognition, and happiness is related to depression and physical activity, we hypothesized that happiness may have an indirect effect on cognition through depression and physical activity aside its direct effect on cognition. In other words, depression and physical activity may play mediatory roles between happiness and cognition. To the best of our knowledge, no study has explored the possible association between happiness and cognition with depression and physical activity as mediatory factors. Therefore, this retrospective cross-sectional study was conducted to assess the direct effect of happiness on cognition and assess the mediatory effects of depression and physical activity on the relationship between happiness and cognition.

### Hypotheses

Briefly, the effect of happiness on cognition is considered as a direct effect, whereas the effect of happiness on cognition through depression or physical activity is considered as an indirect effect. The overall effect of happiness on cognition is considered as the total effect; where total effect = direct effect + indirect effect. Therefore, we proposed four hypotheses to be tested.

Hypothesis 1 (H1): The total effect of happiness on cognition is positively significant.

Hypothesis 2 (H2): The indirect effect of happiness on cognition through depression is significant. In other words, depression plays a mediatory role in the process of happiness and cognition.

Hypothesis 3 (H3): The indirect effect of happiness on cognition through physical activity is significant. In other words, physical activity plays a mediatory role in the process of happiness and cognition.

Hypothesis 4 (H4): The indirect effect of happiness on cognition through depression and physical activity is significant. In other words, depression and physical activity play a joint mediatory role in the process of happiness and cognition.

## MATERIAL AND METHODS

### Data source and study population

The survey data were obtained from the China Family Tracking Survey (CFPS) (http://www.isss.pku.edu.cn/cfps/), which was organized and conducted by the Institute of Social Sciences of Peking University. The CFPS sample was drawn from 25 provinces/municipalities/autonomous regions in China, and they account for 95% of the total Chinese population. Therefore, CPFS can be regarded as a nationally representative sample (*Zhang et al., 2021*). All the sub-sampling frames of CFPS were obtained in three stages: the primary sampling unit consisted of administrative districts/counties; the second stage sampling unit consisted of administrative villages/neighborhood communities; and the third stage sampling unit consisted of households. All the selected households participated in the survey. The baseline survey was conducted in 2010, and follow-up

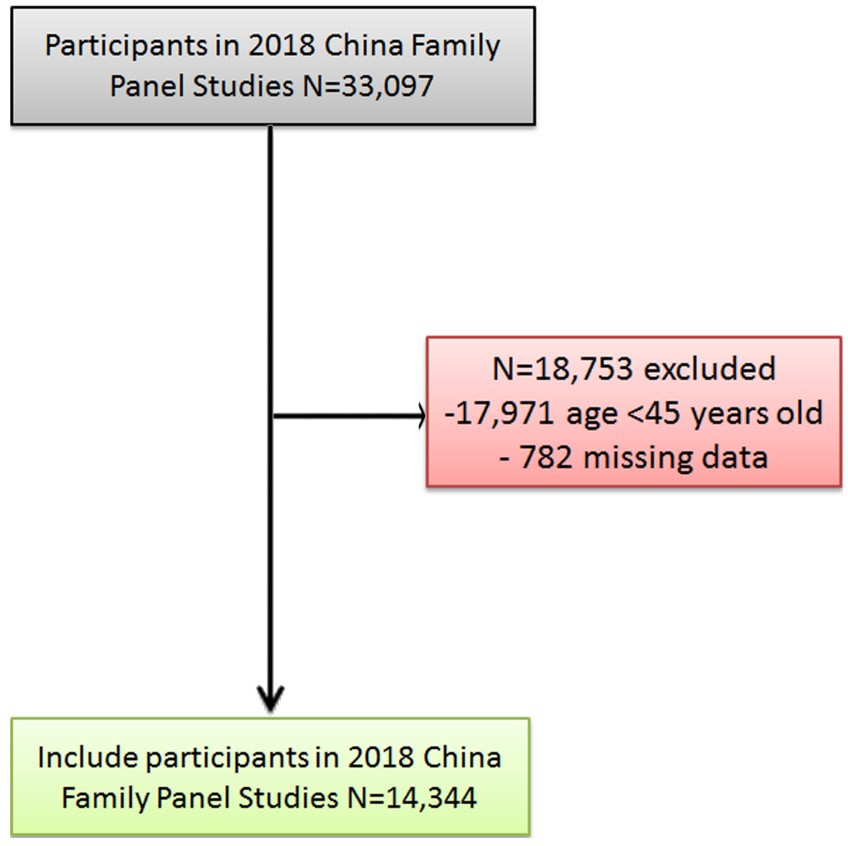

**Figure 1** Flow diagram of the study population.

surveys were conducted biennially. This study was reviewed and approved by the Biomedical Ethics Committee of Peking University (Approval No.: IRB00001052-14010).

In this study, the CFPS 2018 survey data were selected. Of the 33,097 observed participants, 18,753 participants were excluded. Among the excluded group, 17,971 individuals were younger than 45 years, and 782 individuals had missing data for required variables. The final number of participants was 14,344 individuals; these ones had available data on cognitive test and covariates included in this study (Fig. 1).

## Variables
### Dependent variable

This study focused on the association between happiness and cognition. Therefore, cognitive function was considered as the object and set as the dependent variable. Cognitive function was assessed using cognitive modules from the CFPS database. All surveys were conducted in-person but not through telephone calls. Standardized language tests and math tests were included. Each test had 34 questions (*Xie & Lu, 2015*). The questions were derived from the standard curriculum for elementary and middle schools and were arranged in ascending order of difficulty. In the language test, respondents were asked to pronounce words in Mandarin. The mathematics test included a series of mathematical calculation

problems. The first question depended on the educational level of the respondents. If the respondent could not answer the first question or answered incorrectly, the participant was made to answer the question corresponding to the previous educational level until the starting point of the lowest educational level group. If the respondent failed to answer three consecutive questions, answered incorrectly, or answered all questions, the test would be terminated. Therefore, the final test score was determined by the level of the hardest question that each respondent could answer correctly. In this study, the cognition score was the total score on the language and math tests. A higher score indicated a better cognitive function.

### Independent variable

In this study, respondents had to answer questions on how happy they felt. Happiness was set as the independent variable and measured on a 10-point Likert scale, ranging from 1 (very unhappy) to 10 (very happy).

### Mediatory variables

The first mediatory variable was depression, which was assessed using the Center for Epidemiological Research Depression (CES-D) scale. The CES-D scale was developed by the National Institute of Mental Health in 1977, was widely used in epidemiological surveys to screen people with depressive symptoms for further examination and diagnosis. Compared with other self-rating depression scales, CES-D focuses more on individual emotional experience and less on somatic symptoms during depression. The CES-D scale (*Radloff, 1977*) consists of 20 self-reported items and has been widely used in many countries to assess depressive symptoms in the general population (*Kato, 2021*). Higher scores indicated higher degree of depression.

Physical activity was the second mediatory variable. In this study, it was measured by the number of exercises. Respondents recalled the number of exercises in the past week. The exercises included walking, long-distance running, jogging, mountaineering, dancing, yoga, various ball games, water sports (swimming), ice and snow sports, and physical contact sports (judo and boxing).

### Control variables

Since the residence, age, sex, education, smoke, napping, reading, satisfaction of life, income status, and social status of study participants were potential confounders of cognition, they were used as control variables. The definition of control variables was shown in Table 1.

## Statistical analyses

Descriptive and multiple regression analyses of the data were performed using STATA/MP version 16.0. Continuous variables were described using mean ± standard deviation (SD), and categorical variables were described using counts and percentages.

We performed multiple regression analysis using cognitive score as the dependent variable to examine possible associations with various participant demographics and cognitive score. Independent variables were residence, age, sex, marriage, education, smoking, drinking, napping, reading, exercise times, life satisfaction, income status,

**Table 1  Definition of control variables.**

| Control variables | Definition |
|---|---|
| Residence | 1 = urban and 0 = rural |
| Age | The difference between the respondent's date of birth and date of the survey |
| Sex | 1 = man and 0 = woman |
| Education | The highest educational level of respondents |
| Smoke | Whether respondents smoked in the past month, 1 = yes and 0 = no |
| Nap | Whether respondents had the habit of taking a nap, 1= yes and 0 = no |
| Reading | Whether the respondent had read a book in the past year, 1 = yes and 0 = no |
| Life satisfaction | Five levels, with the lowest value of 1 = "very low" and the highest value of 5 = "very high" |
| Income status | Five levels, with the lowest value of 1 = "very low" and the highest value of 5 = "very high" |
| Social status | Five levels, with the lowest value of 1 = "very low" and the highest value of 5 = "very high" |

health status, social status, depression, and happiness. Variables of happiness, depression, life satisfaction, income status, and social status were standardized prior to conducting regression analyses. All independent variables had no multicollinearity (VIF <2). The degree of statistical significance was set at $p < 0.05$.

In this study, conditional process analysis was used to analyze mediatory and moderation effects. It is a method used to test hypotheses about "how" and "under what conditions" a certain factor works (*Hayes & Rockwood, 2019*; *O'Rourke & Mackinnon, 2018*). A theoretical model was created to perform path analysis using the PROCESS macro in IBM SPSS 25 for mediation analyses. The bootstrapping method was used to obtain 95% confidence intervals (95%CI) with 5,000 re-samples for the indirect effects. A mediation analysis (model 6) was performed. An effect was considered significant when the 95% CI did not contain zero.

# RESULTS

## Demographics

The average age of respondents was 59.46 years. Approximately half of the respondents were female (50.08%) and resided in rural areas (52.84%). Most participants had no education (35.63%) or completed primary school (24.85%). Most participants were married (87.62%). More than half of them reported no smoking (69.39%) and drinking habits (82.29%). Most participants had a habit of napping (60.73%). Few participants had recently read books (13.7%). The mean happiness score was 7.51 on a 10-point Likert scale. The average exercise time was 3.08 times in the past week. The average depression score was 33.58, and the average score on the cognitive test was 22.16 (Table 2).

**Table 2  Characteristics of study individuals ($n = 14,344$).**

| Categorical Variables | | N (%) | Continuous variables | $\bar{X} \pm SD$ |
|---|---|---|---|---|
| Sex | Female | 7183 (50.08) | Age (y) | $59.46 \pm 9.715$ |
| | Male | 7161 (49.92) | | |
| Residence | Rural | 7580 (52.84) | Happiness score (1–10) | $7.51 \pm 2.254$ |
| | Urban | 6764 (47.16) | | |
| Education | Illiteracy | 5111 (35.63) | Income level (1–5) | $2.98 \pm 1.134$ |
| | Primary school | 3565 (24.85) | | |
| | Junior high school | 3591 (25.03) | | |
| | Senior high school | 1581 (11.02) | | |
| | Junior college | 339 (2.36) | | |
| | Undergraduate college | 150 (1.05) | | |
| | Master | 7 (0.05) | | |
| Marriage status | Never married | 148 (1.03) | Social status (1–5) | $3.28 \pm 1.118$ |
| | Married | 12568 (87.62) | | |
| | Cohabitation | 70 (0.49) | | |
| | Divorced | 242 (1.69) | | |
| | Widowed | 1316 (9.17) | | |
| Smoking | Yes | 4391 (30.61) | Life satisfaction (1—5) | $4.12 \pm 0.964$ |
| | No | 9953 (69.39) | | |
| Drinking | Yes | 2540 (17.71) | Health status (1–5) | $3.34 \pm 1.241$ |
| | No | 11804 (82.29) | | |
| Nap habit | Yes | 8711 (60.73) | Exercise times | $3.08 \pm 3.55$ |
| | No | 5633 (39.27) | | |
| Reading | Yes | 1965 (13.7) | Depression score | $33.58 \pm 8.554$ |
| | No | 12379 (86.3) | Cognition score | $22.156 \pm 13.951$ |

## Multiple linear regression of cognition

The correlates of cognition were analyzed using multiple linear regression. Residence, age, sex, income level, social status, smoking, napping, reading, education, exercise times, life satisfaction, happiness, and depression had a correlation with cognition (all $p < 0.05$). When other variables were held constant, cognition increased by 0.009 points for each increase in exercise times. Cognition increased by 0.029 SD for every 1 SD increased in happiness and decreased by 0.066 SD for every 1 SD increased in depression (Table 3).

## Mediation analysis
### Total effect of happiness on cognition

The total effect of happiness on cognition was tested. The "total effect" in Table 3 shown the estimated results of the total regression equations when other factors were controlled.

The result showed that happiness had a significant positive total effect on cognition. The total effect value was 0.0496 (95% CI [0.0372~0.0620]). Therefore, Hypothesis 1 was accepted.

**Table 3  Multiple linear regression of cognition.**

| Variables | β | Std. Err. | t | p | 95%CI |
|---|---|---|---|---|---|
| Residence | 0.148 | 0.012 | 12.470 | <0.001 | (0.125, 0.171) |
| Age | −0.014 | 0.001 | −21.130 | <0.001 | (−0.015, −0.012) |
| Sex | 0.231 | 0.015 | 15.710 | <0.001 | (0.202, 0.26) |
| Health status | 0.002 | 0.006 | 0.410 | 0.68 | (−0.009, 0.014) |
| Income level | −0.027 | 0.007 | −4.090 | <0.001 | (−0.04, −0.014) |
| Social status | −0.045 | 0.007 | −6.780 | <0.001 | (−0.058, −0.032) |
| Drinking | −0.002 | 0.016 | −0.150 | 0.88 | (−0.033, 0.029) |
| Smoking | −0.038 | 0.015 | −2.520 | 0.01 | (−0.067, −0.008) |
| Napping | −0.021 | 0.003 | −7.300 | <0.001 | (−0.027, −0.015) |
| Reading | 0.318 | 0.018 | 18.100 | <0.001 | (0.283, 0.352) |
| Education | 0.482 | 0.006 | 86.180 | <0.001 | (0.471, 0.493) |
| Marriage | −0.001 | 0.007 | −0.210 | 0.84 | (−0.014, 0.012) |
| Exercise times | 0.009 | 0.002 | 5.630 | <0.001 | (0.006, 0.012) |
| Life Satisfaction | −0.043 | 0.007 | −6.500 | <0.001 | (−0.056, −0.03) |
| Happiness | 0.029 | 0.007 | 4.390 | <0.001 | (0.016, 0.042) |
| Depression | −0.066 | 0.007 | −10.180 | <0.001 | (−0.079, −0.054) |
| Constant | −0.458 | 0.042 | −10.970 | <0.001 | (−0.54, −0.376) |

## Mediatory role of depression and physical activity

The mediatory effect of depression and physical activity were tested by path analysis. After eliminating the influence of control variables, the direct effect of happiness was statistically significant. The estimate value was 0.0287, which accounted for 57.86% of the total effect of happiness on cognition. The indirect effect of mediators was statistically significant, and the estimate value was 0.0209, which accounted for 42.14% of the total effect. For two mediatory variables, the mediatory effect of depression (path of happiness → depression → cognition) accounted for 38.31% of the total effect. The mediatory effect of physical activity (path of happiness → exercise times → cognition) accounted for 3.02%, and the joint mediatory effect of depression and physical activity (path of happiness → depression → exercise times → cognition) accounted for 0.81%. These results indicated that depression and physical activity had mediatory effects, indicating that happiness had an indirect effect on cognition through depression and physical activity. Therefore, Hypotheses 2, 3, and 4 were all accepted. Mediation analysis results were shown in Tables 4 and 5, and Fig. 2.

## DISCUSSION

The main objective of this study was to explore the association between happiness, depression, physical activity and cognition of the middle and old-aged Chinese individuals. Using CFPS data from 2018, the findings supported our hypothesis that study participants who were happier had higher scores on cognitive tests after controlling for relevant covariates. Moreover, depression and physical activity played mediatory roles between happiness and cognition. Direct effects accounted for 57.86% of the total effects of

**Table 4  Mediatory effect of depression and physical activity in the association between happiness and cognition ($n = 14,344$).**

| Effect | Estimate | BootSE | BootLLCI | BootULCI | Relative effect value (%) |
|---|---|---|---|---|---|
| Total effect (happiness) | 0.0496 | 0.0063 | 0.0372 | 0.0620 | – |
| Direct effect (happiness) | 0.0287 | 0.0066 | 0.0158 | 0.0415 | 57.86 |
| Indirect effect (depression) | 0.0190 | 0.0019 | 0.0154 | 0.0228 | 38.31 |
| Indirect effect (exercise times) | 0.0015 | 0.0014 | 0.0017 | 0.0023 | 3.02 |
| Indirect effect (depression and exercise times) | 0.0014 | 0.0011 | 0.0012 | 0.0016 | 0.81 |
| Total indirect effect of mediators | 0.0209 | 0.0019 | 0.0172 | 0.0248 | 42.14 |

Notes.
BootSE, BootLLCI, BootULCI refer to the standard error, lower limit, and upper limit of 95% confidence interval of indirect effects estimated by the bias corrected percentile Bootstrap method, respectively.

**Table 5  Mediatory model of depression and physical activity between happiness and cognition ($n = 14,344$).**

| Variables | Model 1 | | | | Model 2 | | | | Model 3 | | | |
|---|---|---|---|---|---|---|---|---|---|---|---|---|
| | B | se | t | p | β | se | t | p | β | se | t | p |
| Constant | 0.119 | 0.055 | 2.165 | 0.03 | −0.676 | 0.210 | −3.213 | 0.01 | −0.461 | 0.041 | −11.225 | <0.001 |
| Happiness | −0.290 | 0.008 | −34.324 | <0.001 | 0.161 | 0.034 | 4.783 | <0.001 | 0.029 | 0.007 | 4.375 | <0.001 |
| Depression | – | – | – | – | −0.153 | 0.032 | −4.805 | <0.001 | −0.066 | 0.006 | −10.541 | <0.001 |
| Exercise times | – | – | – | – | – | – | – | – | 0.009 | 0.002 | 5.627 | <0.001 |
| Residence | −0.177 | 0.016 | −11.182 | <0.001 | 0.462 | 0.061 | 7.630 | <0.001 | 0.148 | 0.012 | 12.473 | <0.001 |
| Age | 0.005 | 0.001 | 6.256 | <0.001 | 0.051 | 0.003 | 16.394 | <0.001 | −0.014 | 0.001 | −22.250 | <0.001 |
| Sex | −0.278 | 0.019 | −14.761 | <0.001 | −0.089 | 0.073 | −1.223 | 0.22 | 0.231 | 0.014 | 16.295 | <0.001 |
| Smoking | 0.028 | 0.020 | 1.409 | 0.16 | −0.202 | 0.076 | −2.648 | 0.01 | −0.038 | 0.015 | −2.579 | 0.01 |
| Napping | 0.007 | 0.004 | 1.767 | 0.08 | −0.116 | 0.015 | −7.820 | <0.001 | −0.021 | 0.003 | −7.316 | <0.001 |
| Reading | −0.052 | 0.024 | −2.233 | 0.03 | 0.769 | 0.090 | 8.573 | <0.001 | 0.318 | 0.018 | 18.103 | <0.001 |
| Education | −0.097 | 0.007 | −13.080 | <0.001 | 0.371 | 0.028 | 13.037 | <0.001 | 0.482 | 0.006 | 86.247 | <0.001 |
| Satisfaction | −0.128 | 0.009 | −14.573 | <0.001 | 0.086 | 0.034 | 2.526 | 0.01 | −0.043 | 0.007 | −6.523 | <0.001 |
| Income level | 0.001 | 0.009 | 0.123 | 0.90 | 0.069 | 0.034 | 2.022 | 0.04 | −0.045 | 0.007 | −6.795 | <0.001 |
| Social status | −0.043 | 0.009 | −4.809 | <0.001 | 0.062 | 0.034 | 1.840 | 0.07 | −0.027 | 0.007 | −4.145 | <0.001 |
| $R^2$ | 0.197 | | | | 0.071 | | | | 0.553 | | | |
| F | 318.677 | | | | 91.612 | | | | 1366.076 | | | |

Notes.
Model 1 takes depression as the dependent variable; Model 2 takes exercise times as the dependent variable; Model 3 takes cognition as the dependent variable; In all three models, residence, age, smoking, napping, reading, education, satisfaction, income level and social status related associated with cognition were included into the equation as covariables.

happiness on cognition, and indirect effects accounted for the remaining 42.14% through depression and physical activity.

Mental factors are widely acknowledged to affect physical functioning. Moreover, well-established evidence suggest that mental distress, such as depression, has negative effects on physical health, whereas positive mental state, such as happiness, has beneficial effects on physical health (*Zautra, 2003*). Happiness has substantial effects on psychological and

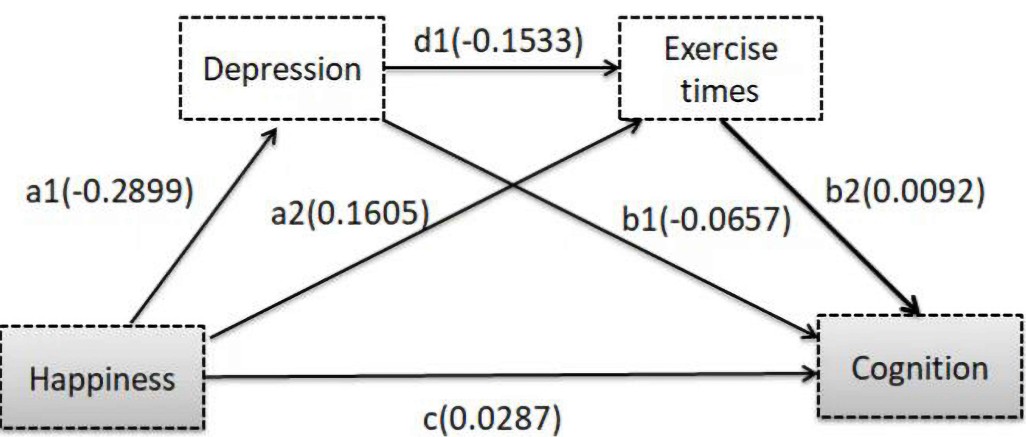

**Figure 2 Depression and exercise times examined as mediators in the association between happiness and cognition.** In this model, the relationship between happiness and cognition was adjusted confounders. The mediation effect of depression on the relationship between happiness and cognition is a1*b1; the mediation effect of exercise times is a2*b2; and the mixed mediation effect of depression and exercise times is a1*d1*b2, while the total effect of happiness on the cognition is a1*b1+a2*b2+a1*d1*b2 +c.

physical health. Currently, happiness is known to foster physical health, but the mechanism underlying this association remains unknown.

Additionally, our study sought to investigate whether happiness can predict cognitive decline in older adults. We found that happiness had a significant positive total effect on cognition. Consistent with our findings, those from studies in the United States and Singapore have shown that individuals with high levels of happiness had a reduced risk of mild cognitive impairment and dementia (*Boyle et al., 2010*; *Zaninotto, Wardle & Steptoe, 2016*). Contrarily, a 12-year follow-up study in Maastricht showed no association between positive emotions and cognitive function among the elderly (*Berk et al., 2017*). Therefore, more evidence is needed to draw firm conclusions. The situation was further complicated by indications that cognitive decline in older adults led to lower happiness levels in the elderly (*Wilson et al., 2013*), implying that the relationship between happiness and cognition was bidirectional.

Regarding the total effect of happiness on cognition, the indirect effect of depression and physical activity on cognition cannot be overemphasized. Evidence suggest that the associations between happiness and depression were complex and may coexist (*Larsen et al., 2017*). Consistent with several previous studies (*Byers & Yaffe, 2011*; *Miskowiak et al., 2012*; *Xu et al., 2015*), this study showed that depression was one of the important risk factors for cognitive decline. Meanwhile, happiness led to reduced risk of depression, and lower depression scores were associated with higher scores of cognition; 38.31% of the total effect between happiness and cognition was mediated by depression. A meta-analysis of longitudinal studies supports our findings that later-life depression increased the risk of cognitive impairment (*Diniz et al., 2013*). Several biologic mechanisms have been observed to be related to the association between depression and mild cognitive impairment. They

include presence of high amyloid plaque number, hypothalamic-pituitary-adrenal axis dysregulation leading to increased glucocorticoid production (*Butters et al., 2008*), and chronic inflammation (*Hermida et al., 2012*). All of these mechanisms may contribute to frontostriatal abnormalities (*Hermida et al., 2012*) or hippocampal atrophy (*Rapp et al., 2006*), resulting in cognitive decline.

In this study, happy individuals had more physical activity levels. Similar to some retrospective studies, this study found physical activity to be one of the protective factors in reducing the risk of cognitive decline (*Ruuskanen & Ruoppila, 1995*; *Stephens, 1988*). Additionally, 3.02% of the total effect between happiness and cognition was mediated by physical activity, implying that happiness was associated with physical activity, and more physical activity in later life confers a protective effect on cognition. Similarly, a systematic review and meta-analysis of longitudinal studies (*Blondell, Hammersley-Mather & Veerman, 2014*) revealed an association between higher levels of physical activity and reduced risk of cognitive decline and dementia. Although the quality of the data is limited, most trial results support the conclusion that moderate levels of physical activity in later life benefit cognitive function in older adults. The cognitive benefits of physical activity include not only maintaining optimal cognitive function but also enhancing existing cognitive function and delaying the onset and progression of cognitive impairment (*Barha et al., 2017*; *Daskalopoulou et al., 2018*; *Almeida et al., 2014*). Furthermore, several studies have reported a dose–response relationship between physical activity and cognitive function, providing additional evidence of the beneficial effect of physical activity on cognitive function (*Gillum & Obisesan, 2010*; *Kasai et al., 2010*; *McAuley et al., 2011*). These studies have suggested a strong relationship between increased physical activity and better cognitive function among older adults. Likewise, several hypotheses exist to explain the beneficial effects of physical activity on cognition. For example, physical activity can promote amyloid clearance and increase cognitive reserve (*Fratiglioni, Paillard-Borg & Winblad, 2004*), brain volume (*Rovio et al., 2010*), and brain-derived neurotrophic factor levels (*Komulainen et al., 2008*). Another trial found that aerobic physical activity can have a direct effect on the body's hippocampal volume, which was beneficial to cognitive function (*Erickson et al., 2011*).

Therefore, ensuring that middle and old-aged individuals in China are happy should not be of utmost importance to society. Strategies to ensure happiness in this population with favorable effects on their mental and physical conditions need to be recognized and developed. Although happiness has a genetic underpinning (*Okbay et al., 2016*), multiple factors can affect it. Tkach and Lyubomirsky (*Tkach & Lyubomirsky, 2006*) summarized eight strategies for increasing happiness; they include mental control, party attendance, affiliation, goal pursuit, religion, inactive hobbies, active leisure, and intentional attempt. These strategies suggest that it is possible to work toward happiness at any age. Effective interventions towards ensuring happiness among middle and old-aged individuals cannot only improve their subjective well-being but also improve cognitive function. Such improvements carry great potential for reducing public health burdens related to cognitive aging and dementia and are necessary interventions needed in a country with an aging population.

## Limitations

Our study had limitations. First, this study was a cross-sectional study with limited power to draw causal inferences from findings. Second, the medical diagnosis of cognitive dysfunction often relied on comprehensive clinical evaluation, and the use of cognitive tests in our study had its limitations. Future studies could consider collecting more clinical data. Despite these limitations, this study, for the first time, used the mediation model to explore the association between happiness and cognitive function among middle and old-aged individuals in China. The large sample size, multivariate analyses controlling for many possible confounds, and the analysis method of conditional process analysis were strengths of the study making it a unique contribution.

## CONCLUSION

This cross-sectional study in a representative population sample suggested that happiness was positively correlated with cognitive function. Mediation analysis further found significant association of depression and physical activity on the relationship between happiness and cognition. This finding provided a better understanding of how happiness has an effect on cognition and provided clues for future research concerned with links between happiness and cognitive function. Effective interventions for happiness of middle and old-aged individuals in China carries great potential for reducing public health burdens related to cognitive aging.

## ACKNOWLEDGEMENTS

We want to give a special acknowledgement to CFPS for providing real and reliable data for academic research.

### Funding

This study was supported by the National Natural Science Foundation of China (Grant No. 82160643). The funders had no role in study design, data collection and analysis, decision to publish, or preparation of the manuscript.

### Grant Disclosures

The following grant information was disclosed by the authors:
National Natural Science Foundation of China: 82160643.

### Competing Interests

The authors declare there are no competing interests.

### Author Contributions

- Xiaojuan Shi analyzed the data, prepared figures and/or tables, and approved the final draft.

- Xiaoxue He analyzed the data, prepared figures and/or tables, and approved the final draft.
- Degong Pan analyzed the data, prepared figures and/or tables, and approved the final draft.
- Hui Qiao conceived and designed the experiments, authored or reviewed drafts of the article, and approved the final draft.
- Jiangping Li conceived and designed the experiments, authored or reviewed drafts of the article, and approved the final draft.

### Human Ethics

The following information was supplied relating to ethical approvals (*i.e.*, approving body and any reference numbers):

All respondents participated in the survey on the basis of informed consent, and this survey has been reviewed by the Biomedical Ethics Committee of Peking University (Approval No.: IRB00001052-14010).

### Ethics

The following information was supplied relating to ethical approvals (*i.e.*, approving body and any reference numbers):

All respondents participated in the survey on the basis of informed consent, and this survey has been reviewed by the Biomedical Ethics Committee of Peking University (Approval No.: IRB00001052-14010).

### Data Availability

The data are available at the China Family Panel Studies (CFPS), a social survey sponsored by the Institute of Social Science Survey (ISSS) of Peking University: http://www.isss.pku.edu.cn/cfps.

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
