# Peer review of "Happiness, depression, physical activity and cognition among the middle and old-aged population in China: a conditional process analysis"

_PeerJ, doi:10.7717/peerj.13673_

## Round 0.1 · original submission · Major Revisions

Thank you for submitting the manuscript to PeerJ. It has been reviewed by experts in the field and we request that you make major revisions before it is processed further.

We look forward to hearing from you soon.

Best wishes,

Badicu Georgian, Ph.D

·

Basic reporting

Authors reported on a conditional process analysis on the happiness, depression, physical activity and cognition among middle and older people in China.

I went through the manuscript and I found that overall, it is suitable for publishing in the Peer J, but I have some major methodological concerns and other issues that the authors need to address before I can accept the manuscript for publication.

Though the paper seems properly organised, it is not and easy to follow. Some issues are not clearly defined or presented (e.g., hypotheses/research questions), which makes it difficult to understand the various parts of the manuscript.

The problem is the lack of proper familiarity with the English language and recommend that the entire article should be corrected by someone who is a professional editor (lines 14-15, 41-42, 48, 53, 56, 58 contains errors and others have no meaning like lines 54-55 or it seems they do not have a common tense, is it present or present perfect; also, lines 61-67 need to be rephrased and the hypothesis needs to be distinctively formulated).

Authors should consider adding more info on background and methods units of the abstract (i.e., questions used in the questionnaire - how many/what kind, procedure they used in the study).

In the entire manuscript the info should be better presented as to raise the interest of the readers. The methods and the results sections look like a string of numbers/percentages, instead they should be presented in an attractive way.

The quality of the images is good enough, but they are quite small. I don’t know if this version has lower resolution than the final version. If not, images should have better resolution in its final size.

Experimental design

The methods seem sufficiently described in this section.
The date of the approval is not given by the authors.
The tense of the used verbs should all be in past (line 93 – “happiness is…”)
Line 96 needs rephrasing, something like The...scale was used to assess depression.
For the control variables it would be better to use a figure/table to illustrate them, maybe using colours or other attractive issues. The paragraph here is somehow boring when it should be presented in an interesting and attractive way to the reader.

Validity of the findings

In Results section, the authors should consider trying to make explanations here less boring and in a better English.

It would be better to have seen more use of terms like 'originality' and 'significance'. Identify what is new in this study that may benefit readers or how it may advance existing knowledge or create new knowledge on this subject. There should be a clear conclusion on why the research findings are significant for health subject and could be used for the help of people in this situation.

Additional comments

Thank you for the opportunity to read your work. This is an interesting topic that can be considered by readers. Nevertheless, there are some concerns with the present manuscript that would need to be addressed for the paper to be able to achieve its potential.

Though the paper is properly organised, it is not and easy to follow. Some issues are not clearly defined or presented (e.g., hypotheses/research questions, methods, results), which makes it difficult to understand the various parts of the manuscript.

·

Basic reporting

The idea of the article is good, but the research is sloppy by the authors. The article needs a major overhaul. Beyond the information in the content of the article, the general framework should be reconstructed. I would like to review the article again after the authors have carefully reconsidered the research.

The introduction should be rewritten. The material and method are sloppy. Completely rewrite. Rewrite Table 1.

Experimental design

Experimental design needs to be explained in more detail.

Validity of the findings

Findings need to be rearranged.

Reviewer 3 ·

Basic reporting

The article is written in english satisfactory and it has sufficient studied literature.
I consider a more rigorous explanation of figures and tables is welcome.

Experimental design

In my opinion the experimental design is within normal limits.
I suggest just a little bit accurate description to the questionaire.

Validity of the findings

Validity is ok. Like I said before is an interesting result of correlation beetween happiness, depression and physical activities, in this country.

---

## Round 0.2 · accepted · Accept

Thank you for submitting the manuscript to PeerJ. Great improvements were performed in the manuscript. Currently, the article is acceptable for publication.

We look forward to hearing from you soon.

Best wishes,

Badicu Georgian, Ph.D

·

Basic reporting

Thank you for providing this comprehensive work.
The authors have presented an improved version of the manuscript.

The introduction provides a proper background of the topic. The manuscript is well-structured, and it is easy to follow the sections.
The quality of the images is good enough, but I don’t know if the reviewing version has lower resolution than the final version. If not, images should have better resolution in its final size.
It seems that the English is technically correct.

I can see that the key words are missing.

Experimental design

The experimental design meets the scope of the journal, and it is relevant to the community.
Methods are described detailed enough.

Validity of the findings

The results and the conclusions are quite interesting and well-discussed. All data are provided.

The authors have adequately addressed all my comments. I have no further suggestions.

·

Basic reporting

First of all, congratulations to the authors. The research idea and design is nice. The research design looks good.

Introduction, method, statistical analysis and discussion of the findings in the research were done very well. The results of the research provide an innovative perspective by contributing to the literature.

I think that the quality and content of the pictures in the text should be improved

Experimental design

Really good.

Validity of the findings

Really good.